# MVLight: Relightable Text-to-3D Generation via Light-conditioned Multi-View Diffusion

## Abstract

Recent advancements in text-to-3D generation, building on the success of high-performance text-to-image generative models, have made it possible to create imaginative and richly textured 3D objects from textual descriptions. However, a key challenge remains in effectively decoupling light-independent and lighting-dependent components to enhance the quality of generated 3D models and their relighting performance. In this paper, we present MVLight, a novel light-conditioned multi-view diffusion model that explicitly integrates lighting conditions directly into the generation process. This enables the model to synthesize high-quality images that faithfully reflect the specified lighting environment across multiple camera views. By leveraging this capability to Score Distillation Sampling (SDS), we can effectively synthesize 3D models with improved geometric precision and relighting capabilities. We validate the effectiveness of MVLight through extensive experiments and a user study. Demo videos: sites.google.com/view/iclr2025-7780/

## 1 Introduction

The evolution of 3D content generation marks a significant advancement in the modern entertainment industry, including virtual reality (VR), animation, video games, and so on. Traditionally, creating a single 3D asset is a labor-intensive process, requiring hours or even days of meticulous work from professional designers. Thanks to the recent breakthroughs in 3D generation models, driven by textual prompts, non-professional users now enable to effortlessly create their own 3D models without a background in art or computer graphics.

A seminal work, DreamFusion (Poole et al., 2023), pioneered text-to-3D generation by harnessing high-performance text-to-image diffusion models such as Stable Diffusion (SD) (Rombach et al., 2022), along with a novel Score Distillation Sampling (SDS) strategy. This approach bypasses the need for direct 3D generation models and synthesizes high-quality 3D models that accurately reflect the textual prompts. However, since text-to-image diffusion models are primarily designed for 2D applications, they lack the necessary geometric multi-view understanding for consistent 3D representation. This limitation often leads to issues such as the Janus multi-face problem or content inconsistencies across different camera views. To address these challenges, recent studies have introduced multi-view diffusion models, e.g., MVDream (Shi et al., 2024), which refine text-to-2D models using 3D self-attention module and multi-view datasets.

Another critical aspect in 3D modeling is the ability to generate relightable models that adapt to varying light conditions. For instance, a rendered 3D object viewed under noon sunlight should be visually different from one observed in the evening. Existing relightable 3D generation methods usually employ Physically Based Rendering (PBR) materials—albedo, roughness, and metallic properties—to achieve this effect. These materials allow for the decoupling of light-independent and light-dependent components, enabling post-generation lighting adjustments. Typically, these models are firstly trained on geometry and albedo properties using multi-view diffusion models, and subsequently fine-tuned on roughness and metallic properties using additional text-to-image models like Stable Diffusion v2 (Rombach et al., 2022). Here, the challenge arises because multi-view diffusion models and text-to-image models typically do not specify lighting information. Consequently, when estimating PBR materials, the absence of direct lighting data complicates the task of

decoupling light-independent and light-dependent components, which in turn adversely affects the quality of relighting effects.

In this paper, we introduce a novel light-conditioned multi-view diffusion model, *MVLight*, which is designed to generate multi-view consistent images under specified lighting conditions. Also, to improve geometric precision and material decomposition, MVLight can additionally generate multi-view consistent normal maps and albedo images. By distilling these properties into SDS, we enable robust and stable training scenarios, as multiple images of an object under varying lighting conditions and camera views can be synthesized to represent the same object. Moreover, unlike most of existing relightable text-to-3D models that rely on Stable Diffusion for PBR material estimation using randomly selected lighting environment, MVLight enables exploiting the same lighting environment for PBR and diffusion outputs for SDS, resulting in more precise disentanglement of light-dependent and light-independent components. We demonstrate the effectiveness of MVLight through comprehensive quantitative and qualitative experiments compared to existing relightable text-to-3D generative models..

In short, out contributions can be summarized as:

- We firstly propose a novel light-conditioned multi-view diffusion model, MVLight, which ensures consistent geometry and specified lighting properties across all output views.
- We validate that the multi-view and light-aware SDS from MVLight improves both the geometric fidelity and relighting capability of the generated 3D models.
- We demonstrate the superiority of MVLight over existing methods via extensive experiments and user study.

## 2 RELATED WORK

In this section, we review the previous literature on text-to-3D generative models, with a specific focus on multi-view and relightable approaches. Also, we highlight the key differences between existing methods and our proposed MVLight.

### 2.1 TEXT-TO-3D GENERATION USING 2D DIFFUSION MODELS

Recent advances in text-to-image generation have achieved remarkable success, largely due to the availability of large-scale datasets such as LAION-5B (Schuhmann et al., 2022), which pair images with textual descriptions, and vision-language models like CLIP (Radford et al., 2021). This success has opened new possibilities for 3D generation by leveraging the strong generative performance of 2D diffusion models. Pioneering works such as DreamFusion (Poole et al., 2023) and SJC (Wang et al., 2023a) introduced the Score Distillation Sampling (SDS) strategy, which distills knowledge from powerful text-to-image models to generate 3D assets by optimizing representations like Neural Radiance Fields (NeRF) (Mildenhall et al., 2020; Wang et al., 2023b; Shi et al., 2024), DMTet (Shen et al., 2021; Lin et al., 2023; Chen et al., 2023; Liu et al., 2023), or Gaussian Splatting (Kerbl et al., 2023; Yi et al., 2024; Liang et al., 2024). Subsequent approaches have focused on enhancing output quality by refining the optimization process, for example, through re-designing the sampling schedule (Tang et al., 2023) or proposing novel loss functions to replace SDS (Liang et al., 2024; Wang et al., 2023b; Huang et al., 2024). However, the primary limitation of these 2D-lifting approaches lies in their reliance on 2D models, which lack 3D consistency across different camera views. This results in sub-optimal 3D models, manifesting issues like the Janus problem or content drift.

### 2.2 MULTI-VIEW DIFFUSION MODELS FOR TEXT-TO-3D GENERATION

To address these limitations, MVDream (Shi et al., 2024) introduced a novel multi-view diffusion model, which fine-tunes Stable Diffusion 2 (Rombach et al., 2022) to generate consistent multi-view images instead of single-view outputs. This was achieved by replacing the self-attention mechanism in the denosing U-Net of the diffusion model with multi-view attention, allowing multiple views to share knowledge and maintain 3D consistency. By leveraging this 3D-aware diffusion prior to SDS, text-to-3D generation model has significantly improved multi-view coherence in rendered 3D models. Following, RichDreamer (Qiu et al., 2024) and UniDream (Liu et al., 2023) propose multi-view

diffusion models that go beyond generating multi-view RGB images. RichDreamer incorporates multi-view depth and normal maps, while UniDream introduces albedo and normal maps to distill strong geometric priors during 3D optimization. These models demonstrate improved geometric fidelity and visual consistency across multiple views.

## 2.3 Relightable 3D Generation

An additional challenge in 3D model synthesis is to generate relightable models that can adapt to varying lighting conditions, offering more natural and realistic renderings. Recent works have tackled this by conditioning NeRF on lighting conditions, allowing the output of the MLP to be different based on the specified lighting (Zhao et al., 2024) for relighting 3D outputs without inverse rendering. Others have adopted Physically Based Rendering (PBR) materials—such as albedo, metallic, and roughness—instead of directly estimating RGB values, enabling models to be relightable under any lighting environment (Xu et al., 2023; Jin et al., 2024).

When it comes to text-to-3D generation, Fantasia3D (Chen et al., 2023) was the first to integrate a PBR pipeline with SDS, producing detailed 3D objects with realistic surface materials. Following, RichDreamer (Qiu et al., 2024) optimizes the geometric properties of 3D models using multi-view normal-depth diffusion models and then estimate albedo, roughness, and metallic properties via Stable Diffusion (SD) 2 (Rombach et al., 2022). Similarly, UniDream (Liu et al., 2023) jointly optimizes the geometry and albedo of the 3D model with multi-view normal-albedo diffusion models while fine-tuning the metallic and roughness components with SD 2. However, both methods have significant limitations. First, they require at least two diffusion models to generate relightable 3D models (one multi-view diffusion models for geometry and one SD 2 for PBR estimation). Second, because their diffusion models lack access to lighting conditions during training, they must blindly estimate the lighting environment while decoupling albedo, metallic, and roughness values, leading to sub-optimal results.

In this paper, we propose MVLight, a novel approach that synthesizes multi-modal outputs—normal maps, albedo, and rendered color—within a single network. Unlike previous methods, MVLight explicitly incorporates lighting information as an input, ensuring that the output multi-view images accurately reflect the specified lighting conditions. This allows for more effective decoupling of albedo, metallic, and roughness values, as the model has direct access to the light environment during optimizing PBR materials, leading to superior relighting performance compared to existing methods.

## 3 Method

In this section, we introduce 1) the architecture of MVLight, a novel light-conditioned multi-view diffusion model that synthesizes 3D-consistent outputs while faithfully adhering to specified lighting conditions across multiple camera views, and 2) how we distill these multi-view outputs along with Ground Truth (GT) lighting information into Score Distillation Sampling (SDS) for relightable text-to-3D generation, improving both geometric fidelity and relighting capabilities. The overview of MVLight and light-aware multi-view SDS are depicted in Fig. 1 and Fig. 2 respectively.

### 3.1 Light-conditioned Multi-view Diffusion Model

**Multi-view consistency.** MVLight extends the U-Net architecture of Stable Diffusion 2 (SD 2) (Rombach et al., 2022), specifically adapted for multi-view generation. Unlike SD 2 which aims to produce 2D image outputs, MVLight generates 4D tensors $\mathbf{x}_0 \in \mathbb{R}^{N_c \times H \times W \times C}$, where $N_c$ is the number of camera views, and $H$, $W$, and $C$ correspond to the spatial dimensions and latent space channels in the VAE latent space (Kingma, 2013). To ensure multi-view consistency, we follow the approach from MVDream (Shi et al., 2024), modifying the self-attention mechanism by reshaping the input from $N_c \times H \times W \times C$ to $(N_c H W) \times C$. This enables the model to learn not only pixel-level correlations within a single view but also across different camera views, thus capturing 3D geometric consistency. Camera-specific information is integrated by injecting extrinsic camera parameters $\zeta \in \mathbb{R}^{N_c \times 16}$ via a 2-layer MLP, which is combined with the diffusion time embeddings. This allows the model to understand the spatial relationships between different camera viewpoints, improving geometric accuracy. Text prompts $\mathbf{y}$ are encoded using the CLIP text encoder (Radford

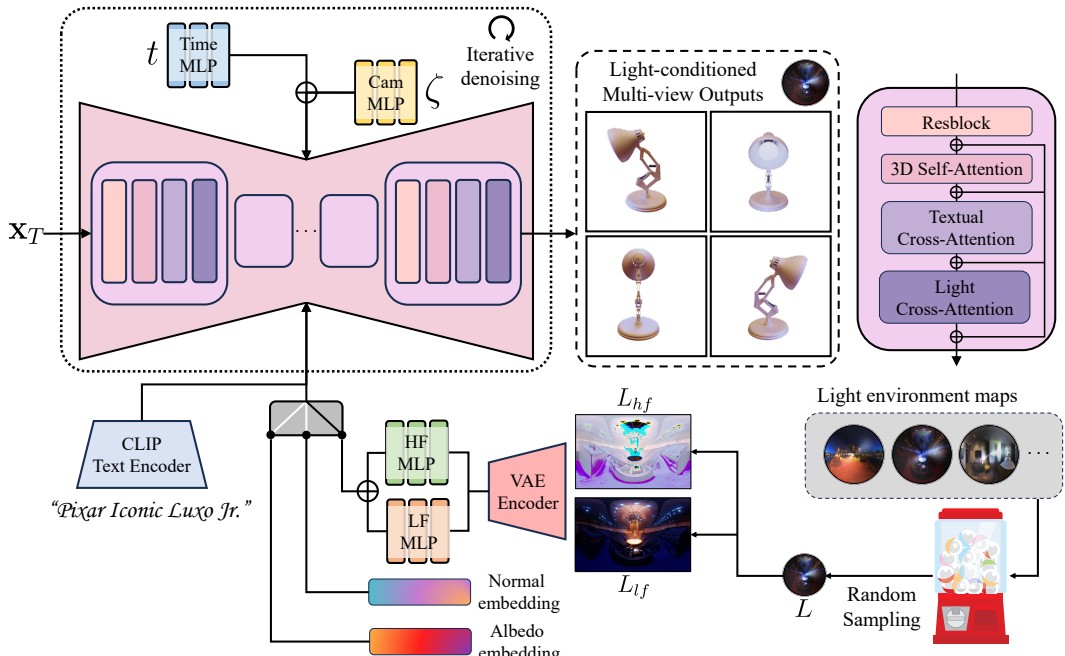

Figure 1: MVLight Overview. MVLight synthesizes 3D-consistent outputs that appear as if captured under specific lighting conditions, with the light environment provided as input across all camera views. Additionally, MVLight generates three distinct modalities—normal, albedo, and RGB—under the specified lighting conditions, enhancing both geometric accuracy and relighting capabilities. Here, $\mathbf{x}_T$ represents random noise input for the diffusion model, $t$ and $\zeta$ denote denoising timestep and camera poses respectively. $L$ refers to the HDR map, with $L_{hf}$ and $L_{lf}$ indicating its high-frequency and low-frequency components, respectively.

et al., 2021) and injected through a textual cross-attention mechanism to ensure that the generated multi-view images align semantically with the conditioned text descriptions.

**HDR-based lighting condition.** MVLight goes beyond textual conditioning and applies lighting conditions directly into the diffusion model. The lighting condition is provided through High Dynamic Range (HDR) images, $L \in \mathbb{R}^{H \times W \times 3}$, where pixel values can range from $[0, \infty)$ to represent light intensity. To embed HDR images effectively for the diffusion model, we follow the approach of Jin et al. (2024) and decouple the HDR image into high-frequency ($L_{hf}$) and low-frequency ($L_{lf}$) components. The high-frequency component captures detailed lighting strength, while the low-frequency component represents the overall color distribution. We compute $L_{hf}$ with non-linear logarithmic mapping and $L_{lf}$ by standard tone-mapping (Debevec et al., 2004; Hasinoff et al., 2016). Both components are reduced in dimensionality via the VAE from SD 2, then flattened and processed by separate 2-layer MLPs. The outputs are combined to form the light embedding $e_l$, which is passed through a new light cross-attention module. This light cross-attention is followed by a textual cross-attention module, forming a sequential residual connection that strongly conditions the diffusion model to both light and text inputs.

In addition to generating light-conditioned images, MVLight is also capable of producing multi-view albedo and normal maps. During training, we introduce learnable embeddings for both normal and albedo, $e_n$ and $e_a$, which share the same dimensionality as the light embedding $e_l$. These embeddings are alternately substituted for light embeddings during training, allowing the model to synthesize outputs among multi-view light-conditioned images, albedo, and normal maps.

**Loss function.** We collect a custom dataset consisting of individual objects or scenes captured from multiple camera viewpoints under $N_l$ distinct lighting conditions, accompanied by corresponding textual descriptions. The details of this dataset, denoted as $\mathcal{X}_{MVL}$, will be further detailed in 4.1. We randomly select light conditions during training from the light environment set, $\mathbb{L} = \{L_a, L_n, L_1, \cdots, L_{N_l}\}$, treating albedo ($L_a$) and normal ($L_n$) as equivalent to lighting conditions. All conditions are assigned equal probability during training: $p = \frac{1}{1+1+N_l}$, ensuring a balanced contribution from albedo, normal, and lighting environments. During training, we jointly uti-

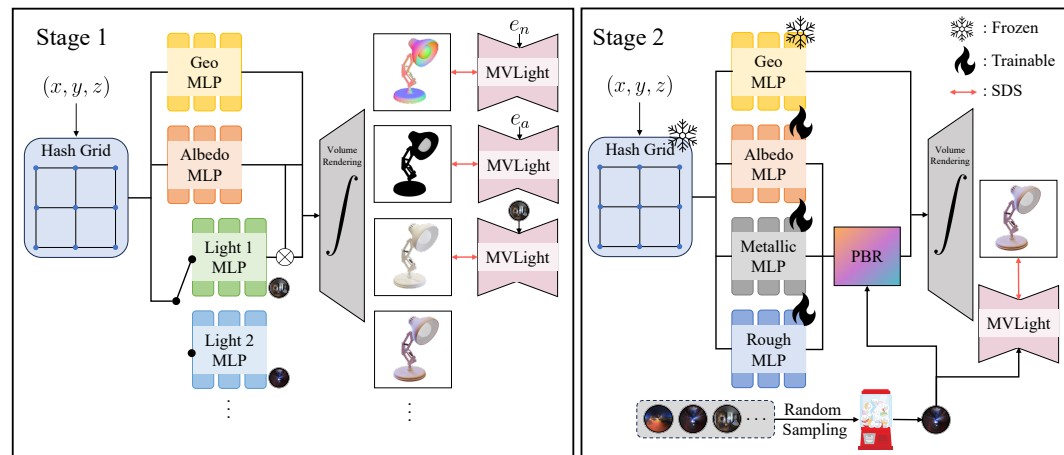

Figure 2: 3D Generation with light-aware multi-view SDS. MVLight is integrated into the SDS optimization pipeline, ensuring 3D consistency while enabling the decomposition of light-dependent and light-independent components. The pipeline consists of two stages: the first focuses on synthesizing the overall geometry and appearance, while the second stage refines PBR materials to enhance relighting capabilities.

lize our synthesized multi-view, multi-light dataset $\mathcal{X}_{MVL}$ along with the large-scale text-to-image dataset $\mathcal{X}$, LAION-5B (Schuhmann et al., 2022), to improve the model's capability to faithfully render 3D outputs based on input text prompts. Training samples $\{\mathbf{x}, \mathbf{y}, \zeta, \mathbb{L}\} \in \mathcal{X}_{MVL} \cup \mathcal{X}$ are used, where $\zeta$ and $\mathbb{L}$ are empty for $\mathcal{X}$. Formally, the light-conditioned multi-view diffusion loss $\mathcal{L}_{MVL}$ is formulated as

$$\mathcal{L}_{MVL}(\theta, \mathcal{X}_{MVL}, \mathcal{X}) = \mathbb{E}_{\mathbf{x}, \mathbf{y}, \zeta, \mathbb{L}, t, \epsilon}\left[||\epsilon - \epsilon_\theta(\mathbf{x}_t; \mathbf{y}, \zeta, e)||\right], \quad (1)$$

$$\text{where } e = \begin{cases} e_a & \text{with probability } \frac{1}{1+1+N_l} \\ e_n & \text{with probability } \frac{1}{1+1+N_l} \\ e_l & \text{with probability } \frac{N_l}{1+1+N_l}. \end{cases}$$

Here, $t$, $\mathbf{x}_t$, and $\epsilon_\theta$ indicate the diffusion timestep, the noisy latent input, and the MVLight model output, respectively. Following MVDream (Shi et al., 2024), during 30% of training iterations, we train MVLight as a 2D text-to-image model using $\mathcal{X}$, skipping 3D self-attention, camera embeddings, and light cross-attention for improved model's flexibility and robustness in text-conditioned image generation.

## 3.2 Light-aware Multi-view guided SDS

We leverage our light-conditioned multi-view diffusion models, MVLight, to synthesize 3D models with Score Distillation Sampling (SDS) (Poole et al., 2023) and Neural Radiance Fields (NeRF) (Mildenhall et al., 2020). Our optimization process for 3D generation is divided into two stages. First, we synthesize the overall geometry and appearance, and then we fine-tune the Physically Based Rendering (PBR) materials—specifically albedo, roughness, and metallic properties—for better relighting performance.

**Geometry and apperance optimization.** In the first stage, similar to other SDS-based approaches (Poole et al., 2023; Shi et al., 2024), we optimize a hash grid-based explicit representation (Müller et al., 2022) of the 3D space, along with a geometry MLP to estimate density $\sigma$ and a color MLP. However, we introduce two key differences: (1) instead of a single color MLP, we use multiple MLPs, each corresponding to a different lighting color distribution; (2) rather than directly estimating shaded RGB values, we use a dedicated MLP to predict albedo and multiple MLPs to compute ambient light under different lighting conditions. This is analogous to a simplified PBR model, where the lighting intensity is assumed to be uniform across the object's surface for each lighting environment. The final RGB color is obtained by multiplying the ambient light values from the lighting MLPs with the albedo from the albedo MLP.

During training, we randomly sample lighting conditions in each SDS iteration. Unlike the diffusion model training, where albedo and normal maps are treated as lighting conditions, we jointly optimize three modalities—normal, albedo, and RGB—during SDS to improve geometric accuracy and albedo estimation. Following MVDream (Shi et al., 2024), we perform SDS on the reconstructed $\mathbf{x}_0$ rather than on the estimated $\epsilon_\theta$ to prevent the over-saturation issue which is common in SDS-based optimization. The light-conditioned multi-view SDS $\mathcal{L}_{MVL-SDS}$ can be formulated as

$$\mathcal{L}_{MVL-SDS}(\phi, (\mathbf{x}^n, \mathbf{x}^a, \mathbf{x}^l) = g(\phi)) = \mathbb{E}_{t,\mathbf{y},\zeta,\epsilon}\left[||\mathbf{x}^n - \hat{\mathbf{x}}_0^n||_2^2 + ||\mathbf{x}^a - \hat{\mathbf{x}}_0^a||_2^2 + ||\mathbf{x}^l - \hat{\mathbf{x}}_0^l||_2^2\right]. \quad (2)$$

Here, $\phi$ represents NeRF parameters, and $g(\phi)$ denotes the volumetric rendering process. The terms $\mathbf{x}^n$, $\mathbf{x}^a$, and $\mathbf{x}^l$ refer to the rendered normal map, albedo, and final rendered color image, respectively, under pre-determined lighting conditions with NeRF. $\hat{\mathbf{x}}^n$, $\hat{\mathbf{x}}^a$, and $\hat{\mathbf{x}}^l$ represent the normal map, albedo, and RGB image predicted by our light-conditioned multi-view diffusion model, with their respective lighting encodings, $\epsilon_\theta(\mathbf{x}_t^n; \mathbf{y}, \zeta, e_n)$, $\epsilon_\theta(\mathbf{x}_t^a; \mathbf{y}, \zeta, e_a)$, $\epsilon_\theta(\mathbf{x}_t^l; \mathbf{y}, \zeta, e_l)$, respectively.

**Light-aware PBR fine-tuning.** After synthesizing 3D models using SDS and NeRF, we fine-tune them by optimizing Physically Based Rendering (PBR) materials—albedo, roughness, and metallic properties—to improve relighting performance. We build on the geometry and albedo MLPs from the previous NeRF optimization stage and introduce two additional MLPs to predict roughness and metallic values. To preserve the geometric properties learned earlier, we freeze the geometry network and only train the PBR materials during this fine-tuning process. We continue using the SDS loss function from Eq. 2, but restrict the optimization to $\mathbf{x}^l$, which represents the final rendered color outputs under a user-specified lighting environment.

In our method, we initialize the PBR lighting environment with the same HDR map used in the light-conditioned diffusion model. Existing approaches (Chen et al., 2023; Liu et al., 2023; Qiu et al., 2024) rely on Stable Diffusion (SD) to estimate PBR materials with SDS. However, SD does not use explicit lighting conditions, meaning that the lighting environment in its outputs is not controlled or known. Consequently, during PBR material estimation, these methods randomly select an HDR lighting environment, which may not match the implicit lighting in the SD outputs. This randomness creates a mismatch between the lighting conditions assumed during PBR and those generated by SD, leading to what we call blind PBR estimation. This mismatch results in sub-optimal performance when estimating PBR materials and therefore sub-optimal relighting performance. In contrast, our approach ensures consistency by using the same HDR map for both the diffusion model in SDS and PBR material optimization, aligning the lighting conditions across the pipeline. This alignment improves the accuracy of PBR material estimation and relighting performance.

## 4 EXPERIMENT

### 4.1 IMPLEMENTATION DETAILS

**Training data synthesis.** As mentioned earlier, we created a custom dataset, $\mathcal{X}_{MVL}$, to train MV-Light, using the Objaverse dataset (Deitke et al., 2023). Each object was captured from 16 camera angles under $N_l$ randomly sampled lighting environments from a set of 450 HDR images collected online (Poly Haven, 2024), along with normal maps and albedo values. With approximately 90,000 objects and $N_l$ set to 4, this process generated around 8.6 million images in total.

**Diffusion model training.** To train the light-conditioned diffusion model, we implemented MV-Light based on the open-source multi-view diffusion model, MVDream (Shi et al., 2024). We initialized the parameters from MVDream and trained the lighting-specific modules, such as the light cross-attention and light embedding MLPs, using the AdamW optimizer (Loshchilov, 2017) with a learning rate of 1e-4. Afterward, we fine-tuned all parameters with a reduced learning rate of 1e-5. For this process, we utilized 32 A100 GPUs, applying a batch size of 128. Fine-tuning for the lighting-specific modules was conducted over 50,000 iterations, while fine-tuning for all parameters occurred over 10,000 iterations with the reduced learning rate.

**SDS optimization.** For 3D model generation using SDS, we integrated MVLight into the threestudio library (Guo et al., 2023) to optimize NeRF-based 3D representations. The output resolution for 3D models during training was set to 256, with the same CFG scale scheduling as MVDream (Shi et al., 2024). Both geometry and appearance optimization (stage 1) and light-awrae PBR fine-tuning (stage 2) involved random sampling from 5 unseen HDR maps, taking 2 hours each for 12,000 iterations on a single A100 GPU.

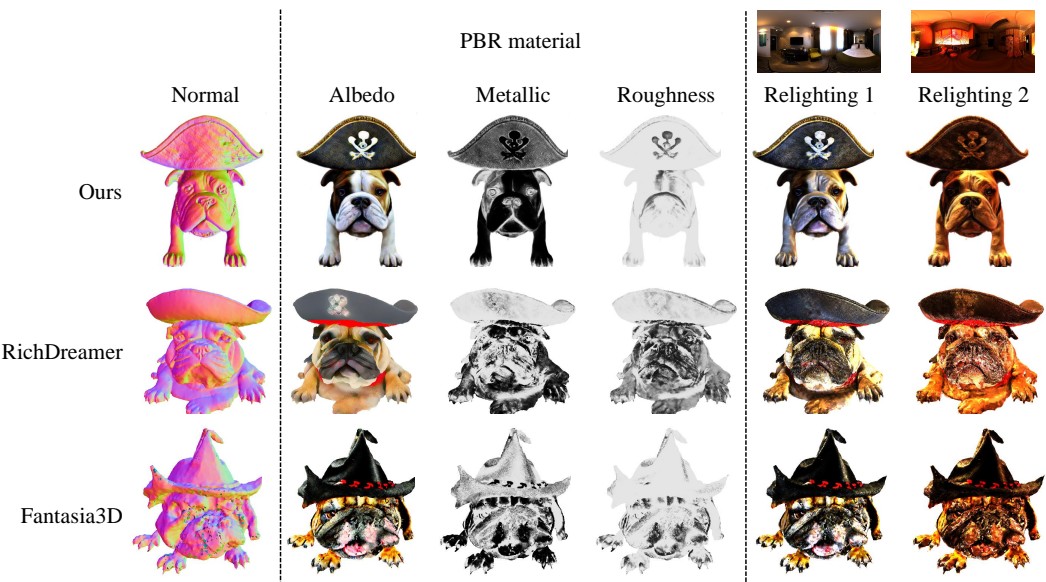

*"A bulldog wearing a black pirate hat"*

Figure 3: Qualitative results of PBR material decomposition and relighted 3D models. Our proposed method enables better estimation of albedo, metallic, and roughness values, which leads to better relighting capability.

Table 1: CLIP score comparison. Our proposed method outperforms existing text-to-3D generative models in a quantitative manner.

|  | DreamFusion | Fantasia3D | MVDream | RichDreamer | Ours |
|---|---|---|---|---|---|
| CLIP Score (↑) | 19.23 | 19.31 | 30.77 | 28.40 | **31.21** |

## 4.2 COMPARISON WITH EXISTING METHODS

**PBR material estimation and relighting capability.** Thanks to MVLight which enables non-blind PBR estimation with light-aware SDS, our method effectively decomposes light-independent and light-dependent components of the 3D object, leading to more accurate PBR material estimation and improved relighting performance. Compared to existing relightable text-to-3D generative models, e.g., Fantasia3D (Chen et al., 2023) and RichDreamer (Qiu et al., 2024), which also utilize PBR-based rendering, MVLight demonstrates superior performance. As shown in Fig. 3, our approach generates more accurate albedo, metallic, and roughness values, whereas existing methods often bake lighting and shadows into the 3D object's appearance due to their blind estimation of lighting conditions. During evaluating relighting performance, we use HDR maps that were never used during the training of both the diffusion models and SDS, allowing us to validate MVLight's generalization to unseen lighting conditions.

**Qualitative results on text-to-3D generation.** We compare MVLight against state-of-the-art (relightable) text-to-3D generation models with available code and checkpoints, which are DreamFusion (Poole et al., 2023), Fantasia3D (Chen et al., 2023), MVDream (Shi et al., 2024), and RichDreamer (Qiu et al., 2024). As shown in Fig. 7, MVLight consistently outperforms competing methods in terms of 3D output quality, exhibiting both stronger faithfulness to input text prompts and superior geometric consistency between RGB and normal outputs. We also provide some 3D video demos here to demonstrate the superiority of MVLight on 3D fidelity in every camera views.

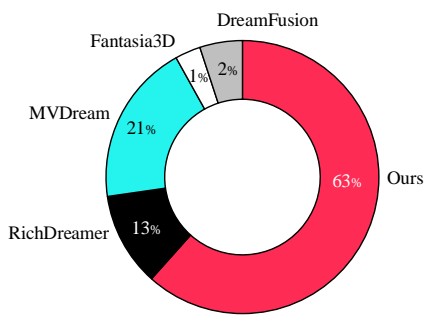

Figure 4: User study.

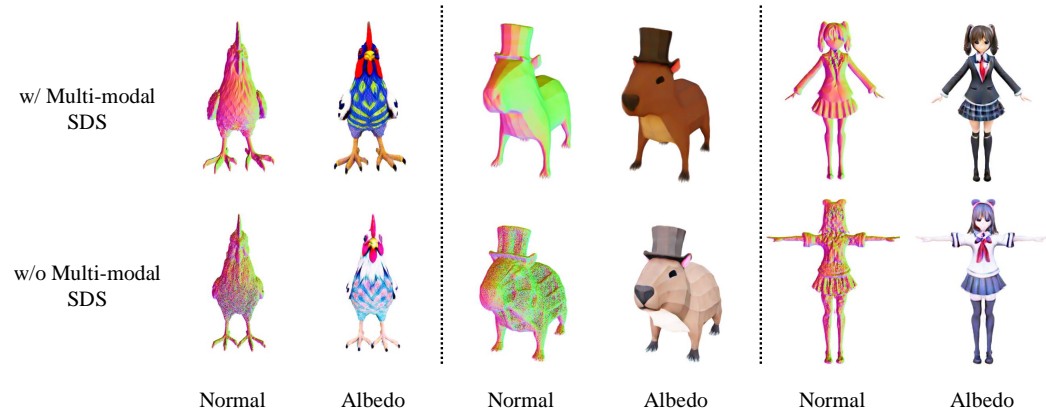

Figure 5: Effectiveness of multi-modal SDS on the estimation of normal map and albedo value. 3D models trained with multi-modal SDS produce smoother normal maps with non-bumpy surfaces and more distinct albedo values with accurate, albedo-like color distribution, compared to those trained without multi-modal SDS.

**CLIP score.** To quantitatively evaluate the performance of MVLight compared to other text-to-3D models, we measured CLIP scores (Radford et al., 2021) to assess how well the generated outputs align with the input text prompts. We used 40 different prompts sourced and modified from Dream-Fusion, MVDream, and newly synthesized ones. CLIP ViT-B/32 model (Dosovitskiy et al., 2020) was employed to extract text and image features, and the CLIP score was computed by averaging the similarity between each view and the corresponding text prompt. As shown in Table 1, MVLight outperforms all competitive methods, demonstrating its ability to generate 3D models that more accurately reflect the input text.

**User study.** Since CLIP scores do not always capture human perception accurately, we conducted a user study to evaluate 40 results generated by each method using corresponding text prompts. The study evaluates geometric texture quality, the realism of 3D model appearances across multiple camera views, and how well they aligned with the text input, highlighting visual differences between the models. As shown in Fig. 4, based on feedback from 24 participants, MVLight emerges as the preferred choice, receiving 63% of the votes. This result highlights the superior visual quality of our approach, validating not only its numerical performance but also its effectiveness in real-world user evaluations.

### 4.3 ABLATION STUDY

**Multi-modal SDS.** When optimizing NeRF-based 3D models for geometric and appearance properties, i.e., Stage 1 in Fig. 2, we utilize multi-modal SDS, which guides the generation of normal, albedo, and rendered color maps through MVLight's view-consistent multi-modal outputs. To validate the effectiveness of multi-modal SDS, we replaced it with single-modal SDS, which only applies supervision to the final rendered color output, similar to Poole et al. (2023); Chen et al. (2023); Shi et al. (2024). As shown in Fig. 5, multi-modal SDS results in 3D models with superior normal maps and albedo values compared to single-modal SDS. Specifically, multi-modal SDS produces smoother normal maps, free from undesired patterns and surface irregularities. For albedo, multi-modal SDS offers more distinct and accurate color representations, as it directly supervises the albedo map. In contrast, single-modal SDS indirectly influences albedo through the final color output, resulting in less accurate, pale, and faint color distributions.

**Light-aware vs blind PBR estimation.** Unlike existing text-to-3D generative models that use Stable Diffusion with randomly sampled HDR light environments for PBR estimation, we utilize our proposed MVLight who produces multi-view outputs under user-specified lighting conditions. By using the same HDR map for PBR that was fed to MVLight during SDS, our method ensures alignment between the lighting environment used for 3D model generation and PBR-based synthesis, leading to more accurate and consistent results. To demonstrate the effectiveness of this light-aware (non-blind) PBR-based 3D model generation, we compare it to blind PBR-based methods, similar

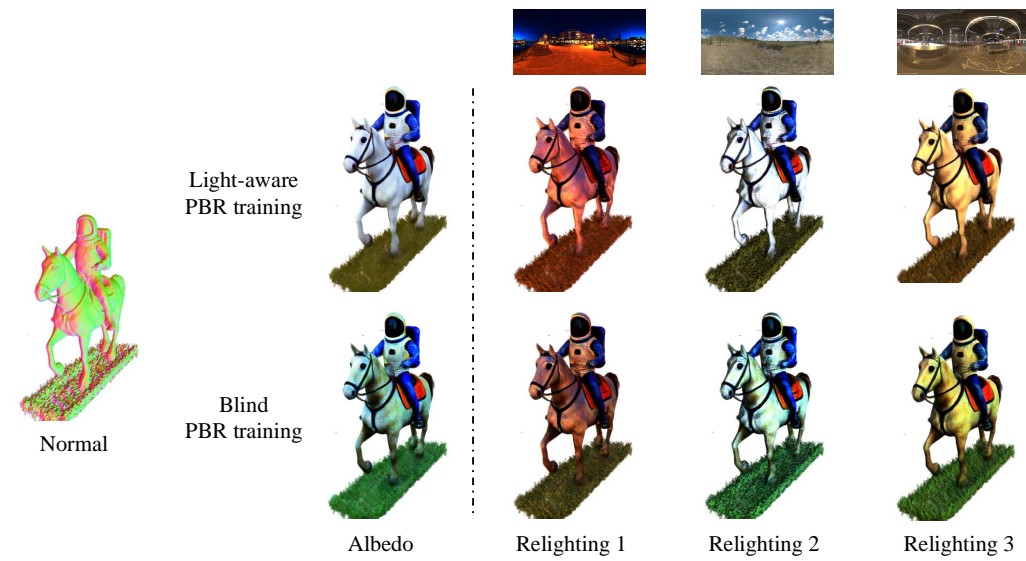

Figure 6: Qualitative relighting results of 3D models using light-aware and blind PBR estimation. The 3D model trained with light-aware PBR exhibits more accurate albedo values, decoupled from light-related components, resulting in superior relighting performance across various HDR lighting environments compared to the model trained with blind PBR.

to Chen et al. (2023); Qiu et al. (2024), which randomly sample a single HDR map during SDS. As shown in Fig. 6, blind PBR-based models often bake lighting and shadows into the albedo values because the lighting conditions used during SDS differ from those used in the PBR synthesis pipeline. This failure to properly decouple light-dependent components from light-independent ones results in sub-optimal albedo estimation and poorer relighting performance under various lighting conditions. In contrast, our light-aware PBR-based 3D model generation better decouples light-dependent components, leading to more accurate albedo values and significantly improved relighting performance across different lighting environments.

## 5 DISCUSSION AND CONCLUSION

**Conclusion.** In this paper, we propose MVLight, the first light-conditioned multi-view diffusion model, capable of generating 3D-consistent outputs under user-specified lighting conditions across all camera views, along with albedo and normal maps. By integrating MVLight into the Score Distillation Sampling (SDS) pipeline, we improve the geometric and appearance fidelity of synthesized 3D models, enabling more accurate decomposition of light-dependent components. Unlike existing text-to-3D generative models that blindly estimate PBR materials by randomly selecting lighting conditions, MVLight supports non-blind PBR estimation, ensuring alignment between the lighting conditions used during PBR and SDS. We demonstrate the effectiveness of MVLight through extensive experiments, showcasing its superior performance in text-to-3D generation, PBR material estimation, and relighting capabilities compared to other state-of-the-art methods.

**Limitation.** While MVLight successfully generates three distinct modalities—albedo, normal, and shaded color—across multiple views, there is no guarantee of alignment between these modalities, as they are synthesized independently. When attempting to train MVLight with modality alignment using self-attention mechanisms, similar to Wonder3D (Long et al., 2024) or UniDream (Liu et al., 2023), we empirically observed a *trade-off* between output quality and modality alignment. In this paper, we prioritize output quality and rely on SDS to manage misalignments between modalities, but this misalignment could lead to sub-optimal performance in certain cases. We address this limitation and leave developing a model architecture that learns modality alignment without compromising output quality as a future work.

DreamFusion Fantasia3D MVDream RichDreamer Ours

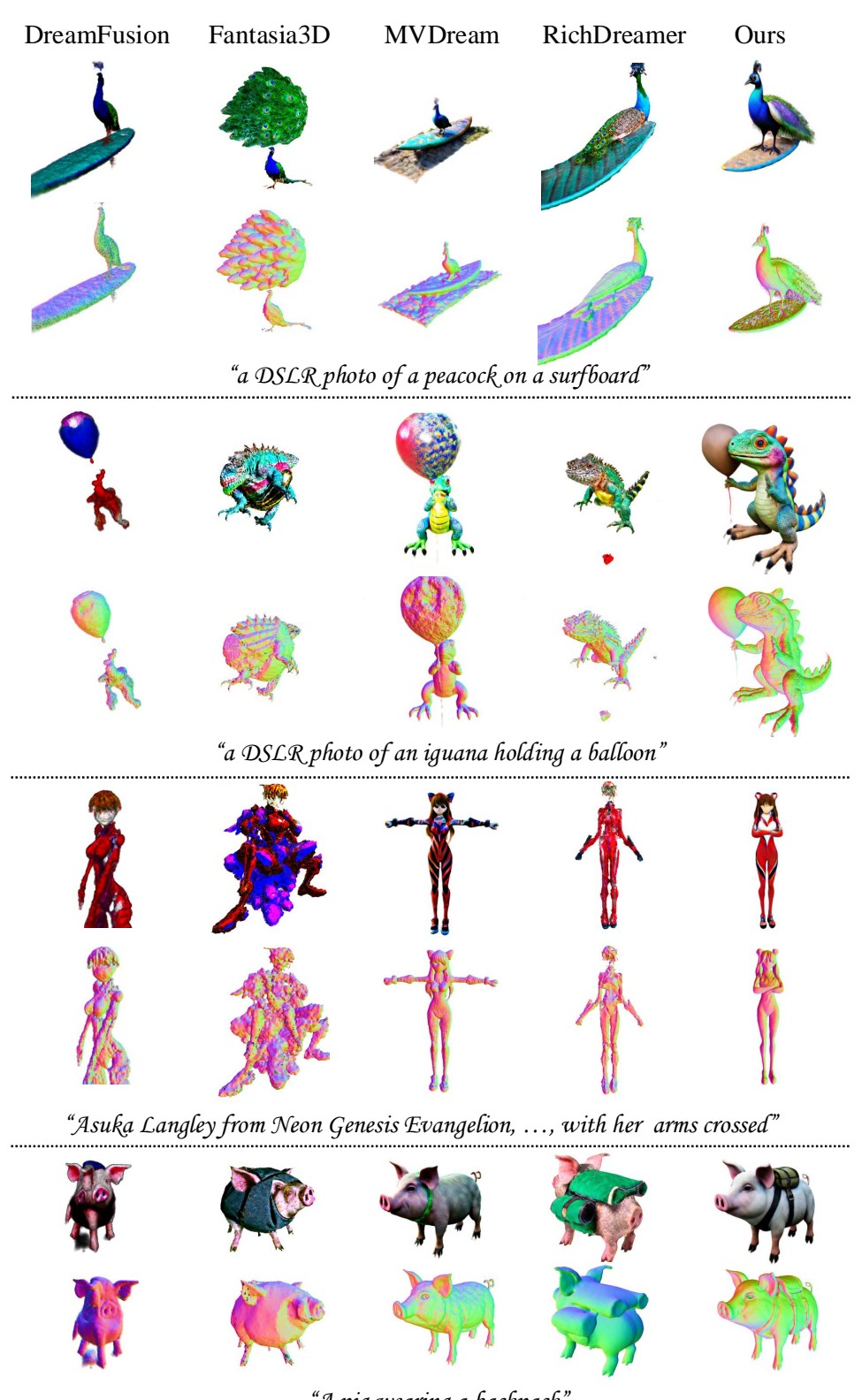

*"a DSLR photo of a peacock on a surfboard"*

*"a DSLR photo of an iguana holding a balloon"*

*"Asuka Langley from Neon Genesis Evangelion, …, with her arms crossed"*

*"A pig wearing a backpack"*

Figure 7: Qualitative results on text-to-3D generation. Our proposed MVLight outperforms all the other comparative methods (Poole et al., 2023; Chen et al., 2023; Shi et al., 2024; Qiu et al., 2024) both in visual quality with their semantically aligned to input text prompt and geometric fidelity that the normal map seems to plausible paired with RGB outputs.

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

## A  ADDITIONAL QUALITATIVE RESULTS

We present additional qualitative comparisons of text-to-3D generation performance, exhibiting our proposed MVLight model alongside other existing (relightable) text-to-3D models, including DreamFusion (Poole et al., 2023), Fantasia3D (Chen et al., 2023), MVDream (Shi et al., 2024), and RichDreamer (Qiu et al., 2024). As demonstrated in Fig. 9 and 10, MVLight consistently outperforms these methods in terms of 3D appearance consistency, geometric fidelity, and the semantic accuracy between the input text prompt and the generated 3D model.

In Fig. 8, we further illustrate MVLight's qualitative performance in PBR material estimation and relighting across multiple views with various text inputs. MVLight excels at decoupling light-dependent features from the 3D model and produces plausible PBR material estimates. Additionally, our model adapts effectively to different and unseen lighting environments, demonstrating robust performance across diverse lighting conditions.

## B  ETHICAL STATEMENT

The light-conditioned multi-view diffusion model, MVLight, proposed in this paper is designed to enhance the 3D generation task, which is particularly relevant in fields such as entertainment, media, and gaming. However, we acknowledge the potential for misuse, including the generation of harmful content such as violent or sexually explicit materials through third-party fine-tuning. Furthermore, since our model builds upon prior works like Stable Diffusion (Rombach et al., 2022) and MVDream (Shi et al., 2024), it may also inherit inherent biases and limitations that could lead to unintended outputs. We emphasize the importance of clearly presenting the outputs from MVLight as synthetic and urge that generated images or 3D models be critically evaluated to prevent misrepresentation. Additionally, while these generative models offer valuable advancements in creative fields, they also raise concerns about the displacement of human labor through automation. However, we believe these tools have the potential to foster innovation and enhance accessibility within the creative industries, supporting new opportunities for growth and collaboration.

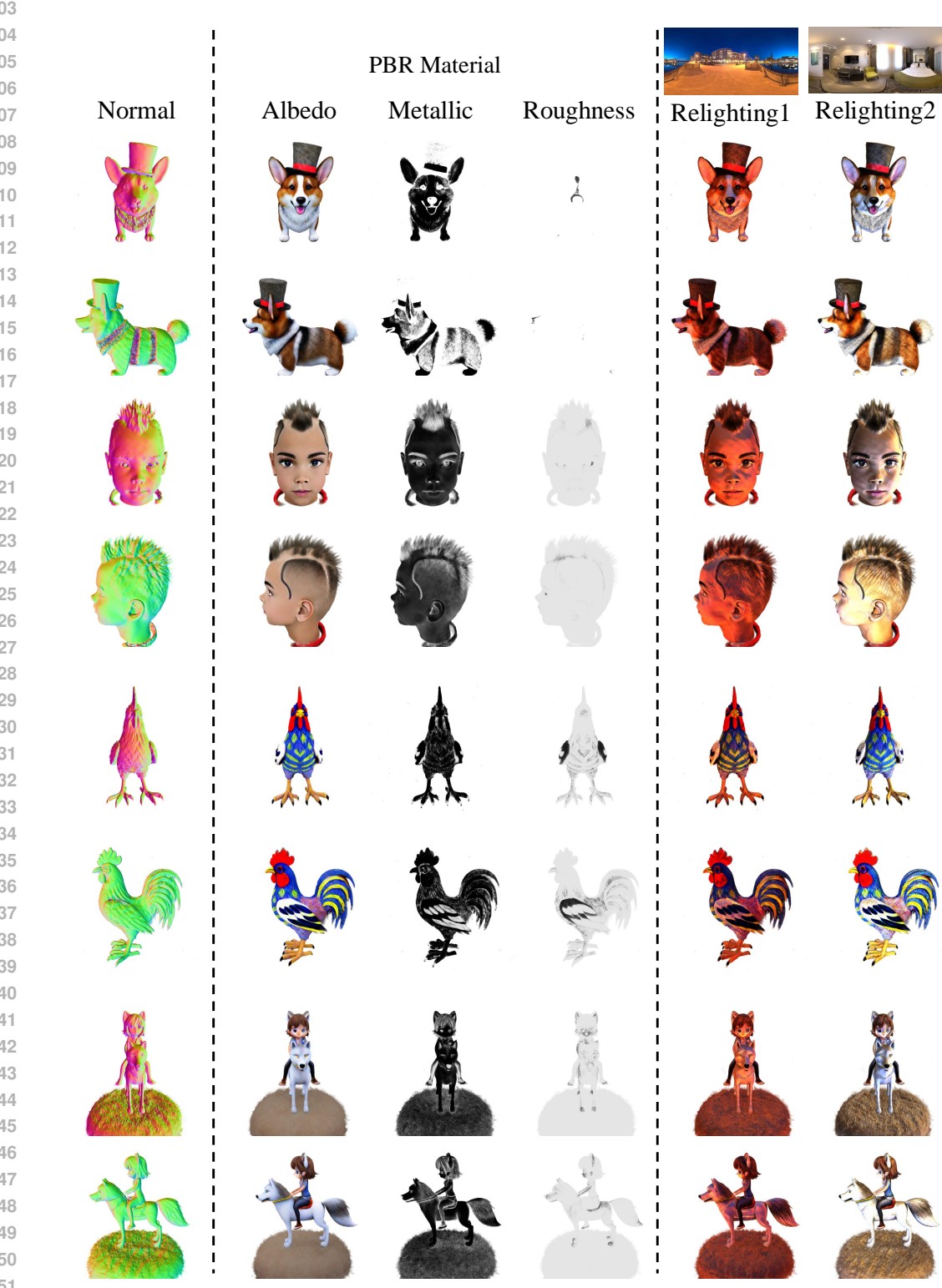

Figure 8: Additional qualitative results on PBR material estimation and relighting performance of MVVLight.

DreamFusion  Fantasia3D  MVDream  RichDreamer  Ours

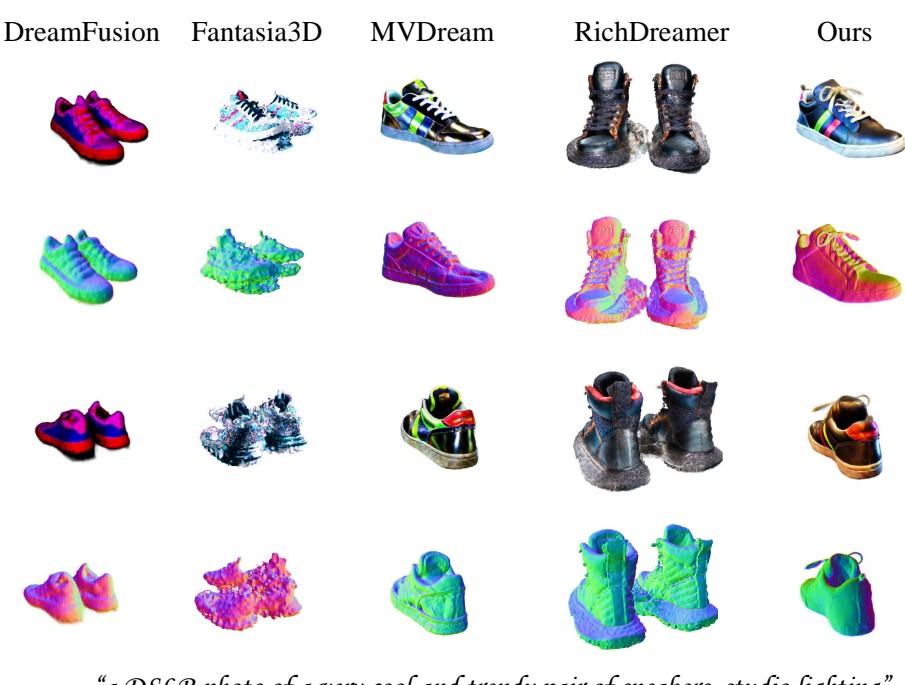

*"a DSLR photo of a very cool and trendy pair of sneakers, studio lighting"*

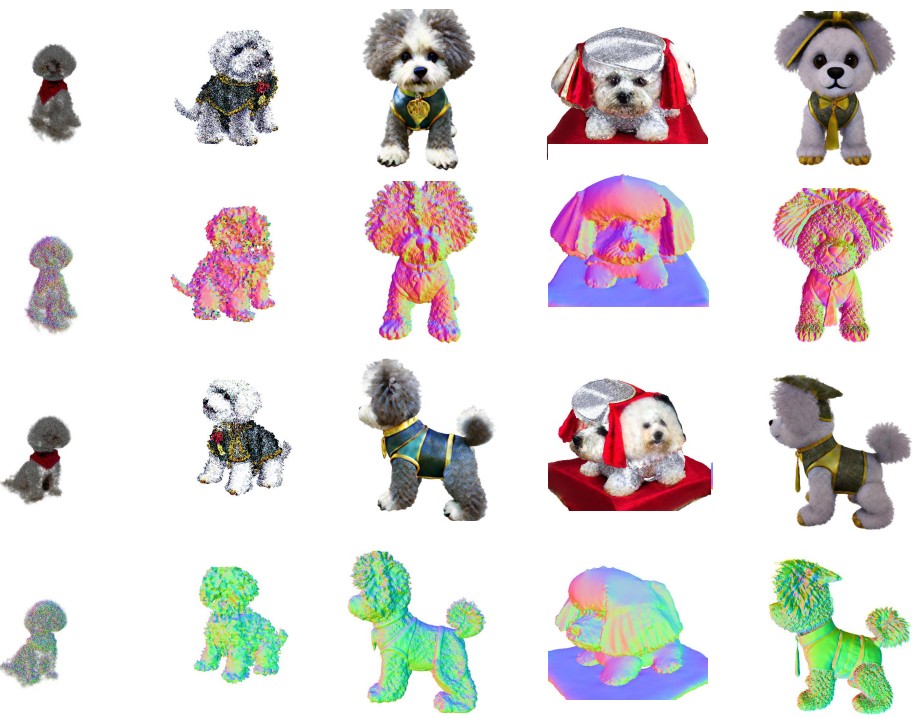

*"a bichon frise wearing academic regalia"*

Figure 9: Additional qualitative results on text-to-3D generation (1).

DreamFusion Fantasia3D MVDream RichDreamer Ours

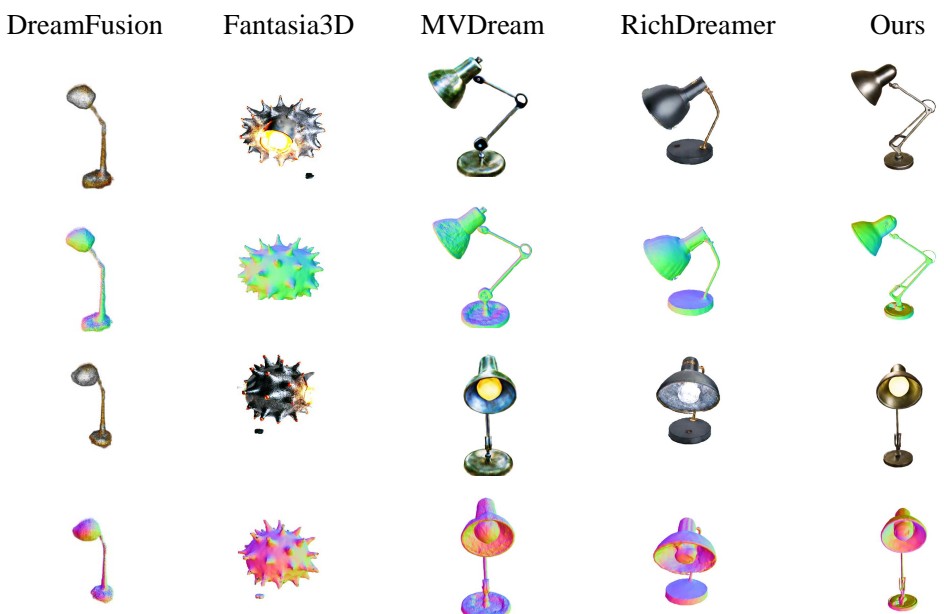

*"Luxo Jr. from Pixar, the small, flexible desk lamp with a sleek metallic body, a round head with a light bulb, and articulated joints"*

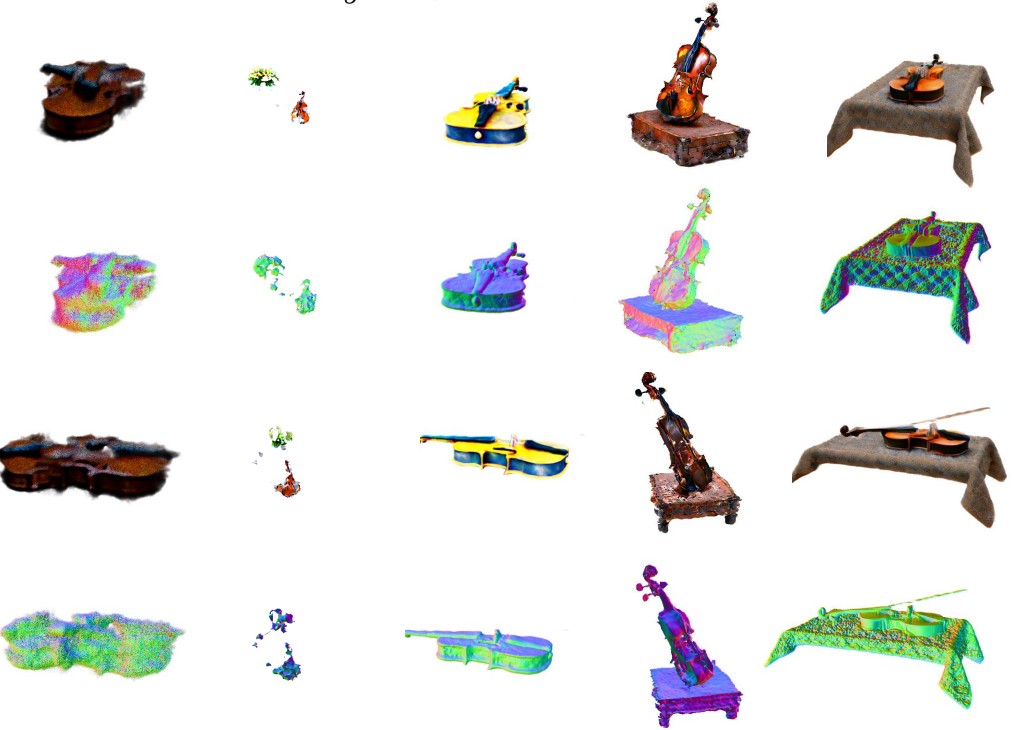

*"a DSLR photo of a beautiful violin sitting flat on a table"*

Figure 10: Additional qualitative results on text-to-3D generation (2).

