# OpenReview forum: "MVLight: Relightable Text-to-3D Generation via Light-conditioned Multi-View Diffusion"
_ICLR.cc/2025/Conference — ICLR 2025 Conference Withdrawn Submission_

### Official Review · Reviewer_kxMk · 2024-10-28

**Soundness:** 2
**Presentation:** 3
**Contribution:** 2
**Rating:** 5
**Confidence:** 4

**Summary:**

The paper addresses the task of relightable object generation, tackling the previous technical challenge of decoupling light-independent and lighting-dependent components in existing methods. The key insight and motivation is the proposal of a light-conditioned multi-view diffusion model that generates high-quality images consistent with specific environmental lighting. Based on this model, the paper optimizes the 3D object's normal and albedo. Subsequently, with the object's geometry and albedo fixed, the paper further optimizes its Metallic and Roughness properties to achieve objects that can be realistically relit under various lighting conditions.

**Strengths:**

1. The experiments and ablation studies are sufficient.
2. The paper is well written and easy to read.

**Weaknesses:**

1. From the results in Figure 7, it appears that the 3D objects generated by the paper suffer from color oversaturation.
2. Figure 6 presents relighting results, but the objects appear to be completely diffuse, lacking any reflective qualities. This makes the objects seem unrealistic. The paper needs to explain why it is unable to synthesize materials with high realism.
3. The albedo synthesized in the results from Figure 8 is not uniform and exhibits oversaturation. The Metallic and Roughness properties also seem incorrect.
4. Personally, I feel that the paper's methodology takes a roundabout approach. The paper first trains a light-conditioned multi-view diffusion model and then optimizes relightable objects based on this multi-view diffusion model and SDS loss. A more direct approach might be to train a multi-view material generation model with these relightable 3D assets to directly generate consistent multi-view albedo, metallic, and roughness, and then project them onto 3D, similar to the one-to-345++ method, to obtain an initial relightable object. This object could then be further optimized based on the SDS loss. I recall that UniDream has a similar approach.
5. The roundabout approach taken by the paper is likely to result in information loss, as the SDS Loss is not a one-to-one corresponding loss function.
6. The paper's comparative experiments and ablation studies are quite comprehensive. My main concern is that the paper's design concept is outdated, and the results are not good enough. I feel that this level of technical achievement should not be accepted by ICLR.

**Questions:**

I mainly care about the poor quality and the outdated design of this paper.

---

> ### Comment · Reviewer_kxMk · 2024-11-26
>
> Since there is no response from authors, I will maintain my score.

---

### Official Review · Reviewer_iUei · 2024-11-01

**Soundness:** 4
**Presentation:** 3
**Contribution:** 3
**Rating:** 6
**Confidence:** 4

**Summary:**

This paper argues that previous text-to-image generative models do not specify lighting information, which will harm the quality of relighting effects. To address this issue, the paper proposes a novel light-conditioned multi-view diffusion model that explicitly integrates lighting conditions into the generation process, enabling the generation of images under varying lighting conditions. Additionally, the paper introduces a two-stage light-aware multi-view SDS algorithm that enhances the modeling of objects' PBR materials, thereby improving their relighting capabilities.

**Strengths:**

1. This paper proposes a light-conditioned multi-view diffusion model that enables the generation of consistent multi-view images under different lighting conditions (different HDR in the paper), which sounds novel.

2. Based on their light-conditioned multi-view diffusion model, the authors propose a light-aware multi-view SDS that optimizes object PBR materials under varying lighting conditions. Compared to the baseline, the paper demonstrates significant improvements in both PBR modeling and relighting capabilities.

3. The pretrained model should benefit further research on relightable object rendering as long as the authors release their checkpoint.

4. The paper is well-written and easy to understand in overall.

**Weaknesses:**

1. Changing different lighting conditions in the light-aware multi-view SDS may require a long optimization time.

2. I understand it is hard to evaluate the quality of relighting results, but the metrics shown in the paper is CLIP Score. The user study is reasonable, but the authors collect response from 24 participants, which does not seem to be a large number.

**Questions:**

Apart from the aforementioned weakness part, I also have the following questions:

1. How to get albedo $L_a$ and normal $L_n$ during training and optimization using the SDS approach?

2. This paper adopts a text-to-3D paradigm, which is not very controllable compared with image-to-3D. Though it is beyond the scope of this paper, does the author have any thoughts in mind about how to extending this framework to more controllable relightable object generation?

3. The required training resources for the diffusion model seem huge, can the authors elaborate on the rough training time?

4. Since the model can achieve consistent optimization results applying SDS, is it possible to extract a UV map for each material, which should be useful for downstream editing tasks?

Overall, I lean towards accepting the paper, and I would be glad to raise my rating if my concerns can be addressed.

---

### Official Review · Reviewer_9qoz · 2024-11-02

**Soundness:** 2
**Presentation:** 3
**Contribution:** 2
**Rating:** 3
**Confidence:** 4

**Summary:**

This paper presents MVLight, a light-conditioned multi-view diffusion model for text-to-3D generation that aims to integrate lighting conditions into the generation/optimization process. The key idea is to enable the model to synthesize 3D-consistent outputs while adhering to specified lighting conditions across multiple camera views. The authors extend the multi-view diffusion architecture by incorporating HDR environment maps as explicit lighting conditions, processing them into high and low-frequency components for better lighting representation. The model is designed to generate not only RGB images but also normal maps and albedo values(?), which are then used in a two-stage optimization process: first for geometry and appearance, then for PBR material properties.

The method leverages Score Distillation Sampling (SDS) with a light-aware approach, where the same lighting environment is used for both diffusion outputs and PBR material optimization. This differs from existing methods that rely on randomly selected lighting environments during PBR estimation. The authors train their model on a custom dataset created from Objaverse, using 450 HDR environment maps, and demonstrate their results through comparisons with existing text-to-3D generation methods like DreamFusion, Fantasia3D, MVDream, and RichDreamer. Their evaluation includes both qualitative results and quantitative metrics using CLIP scores, along with a user study involving 24 participants.

**Strengths:**

- The paper presents an attempt to explicitly integrate lighting conditions into the multi-view diffusion process for text-to-3D generation. The idea of using the same lighting environment for both diffusion outputs and PBR optimization (non-blind PBR estimation) shows thoughtful consideration of the lighting consistency problem.
- The authors make several reasonable architectural choices in extending the multi-view diffusion framework. The two-stage optimization process (geometry/appearance followed by PBR materials) provides a common, structured approach to the problem.
- The paper addresses an important challenge in text-to-3D generation - the need for better control over lighting conditions and more accurate material decomposition.
- The paper provides reasonable implementation details about their training process, including specifics about GPU requirements and optimization parameters.

**Weaknesses:**

- A significant weakness is the paper's failure to acknowledge and compare with several crucial recent works in lighting-controllable generation. The paper fails to address several crucial recent works in lighting-controllable generation, notably DreamMat (SIGGRAPH 2024), DiLightNet (SIGGRAPH 2024), and Neural Gaffer (NeurIPS 2024). This oversight undermines the paper's claimed novelty and contribution. These methods might provide valuable insights or even achieve comparable results in terms of lighting control. This oversight raises questions about the true advancement this work represents in the field.
- The paper employs a simplified ambient lighting setting during optimization (lines 267-269), which is inadequately justified and appears to compromise the method's ability to handle complex lighting scenarios. This limitation appears in the results, which notably lack high-frequency lighting details and realistic specular effects.
- The paper's handling of normal maps and albedo is confusing - Figure 1 shows them as both conditioning embeddings and generation targets, the relationship between these dual roles is unclear. How are these embeddings learned?
- The method's computational requirements are expensive - taking 2 hours on an A100 GPU per asset - without adequate justification for this overhead. This is particularly concerning given that recent methods like SF3D can achieve similar or better results in seconds.
- The user study with only 24 participants is too small for reliable conclusions. The paper lacks lighting-specific metrics for quantitative evaluation, relying heavily on CLIP scores which may not correlate well with lighting quality. There's also no analysis of failure cases with challenging lighting and materials.

**Questions:**

- The paper proposes to use simplified ambient lighting during optimization (lines 267-269), which seems to contradict the goal of high-quality relighting. Could you explain the technical motivation behind this design choice? How does this simplification affect the model's ability to capture complex lighting interactions, especially high-frequency details? Have you conducted experiments with more realistic lighting models during optimization?
- Recent works have shown significant progress in lighting-controllable generation. Could you clarify how MVLight advances beyond these methods?
- The architecture appears to use normal maps and albedo as both conditioning signals and generation targets (Fig. 1). Could you elaborate on this design choice?

---

### Official Review · Reviewer_BNsm · 2024-11-04

**Soundness:** 3
**Presentation:** 3
**Contribution:** 2
**Rating:** 3
**Confidence:** 4

**Summary:**

The paper presents MVLight, an innovative multi-view diffusion model that conditions on environmental lighting while incorporating normal and albedo embeddings. This approach addresses ambiguities in lighting conditions and multi-view output by clarifying these factors. Additionally, the paper introduces a method to distill the 3D model and PBR textures from MVLight using SDS, creating a new text-to-ad generation framework that enhances geometry and produces high-quality PBR materials.

**Strengths:**

1. Generating realistic PBR materials remains challenging due to ambiguities in PBR rendering decomposition. MVLight’s introduction of learnable embeddings in the multi-view training pipeline is a simple yet effective solution to address these ambiguities.
2. The model demonstrates good performance. The generated meshes and albedo are clean, and the user study indicates that this method outperforms competing approaches like RichDreamer and Fantasia3D, etc.

**Weaknesses:**

1. Lack of raw multi-view results: To confirm the accuracy of the multi-view outputs, it would be beneficial for the authors to display raw outputs and analyze how conditioning affects them.
2. Evaluation of raw multi-view results: Comparison with models like Collaborative Control for Geometry-Conditioned PBR Image Generation (ECCV 2024) could further contextualize MVLight’s performance.
3. Limited comparison with other PBR-focused methods: Given that the paper emphasizes 3D generation but estimates PBR materials post-geometry, comparisons with PBR-focused works such as Paint-it (CVPR 2024), DreamMat (SIGGRAPH 2024), and FlashTex (ECCV 2024) would enhance the evaluation.
4. Limited object diversity: The results focus primarily on objects with low metallic content and high roughness (e.g., organic forms). Additional PBR visual comparisons for materials like metal and glass would demonstrate the method’s versatility.

Reference:

[1]. Vainer et al., Collaborative Control for Geometry-Conditioned PBR Image Generation. ECCV. 2024

[2]. Kim et al., Paint-it: Text-to-texture synthesis via deep convolutional texture map optimization and physically-based rendering. CVPR. 2024

[3]. Zhang et al., DreamMat: High-quality PBR Material Generation with Geometry-and Light-aware Diffusion Models. TOG. 2024

[4]. Deng et al., FlashTex: Fast Relightable Mesh Texturing with LightControlNet. ECCV. 2024

**Questions:**

1. How do the albedo and normal embeddings influence the output? What specific function do they serve?
2. Could you clarify the batch size used during training? A size of 128 seems small for a diffusion model; is this per GPU, or is the total batch size 8 x 128?

Please also see the weakness, I will be willing to raise my rating if these concerns are resolved.

---

### Note · Authors · 2024-12-03

I have read and agree with the venue's withdrawal policy on behalf of myself and my co-authors.